# Conditional Networks

## Abstract

In this work we tackle the problem of out-of-distribution generalization through conditional computation. Real-world applications often exhibit a larger distributional shift between training and test data than most datasets used in research. On the other hand, training data in such applications often comes with additional annotation. We propose a method for leveraging this extra information by using an auxiliary network that modulates activations of the main network. We show that this approach improves performance over a strong baseline on the Inria Aerial Image Labeling and the Tumor Infiltrating Lymphocytes (TIL) Datasets, which by design evaluate out-of-distribution generalization in both semantic segmentation and image classification.

## 1 Introduction

Deep learning has achieved great success in many core artificial intelligence (AI) tasks (Hinton et al., 2012; Krizhevsky et al., 2012; Brown et al., 2020) over the past decade. This is often attributed to better computational resources (Brock et al., 2018) and large-scale datasets (Deng et al., 2009).

Collecting and annotating datasets which represent a sufficient diversity of real-world test scenarios for every task or domain is extremely expensive and time-consuming. Hence, sufficient training data may not always be available. Due to many factors of variation (e.g., weather, season, daytime, illumination, view angle, sensor, and image quality), there is often a distributional change or domain shift that can degrade performance in real-world applications (Shimodaira, 2000; Wang & Schneider, 2014; Chung et al., 2018). Applications in remote sensing, medical imaging, and Earth observation commonly suffer from distributional shifts resulting from atmospheric changes, seasonality, weather, use of different scanning sensors, different calibration and other variations which translate to unexpected behavior at test time (Zhu et al., 2017; Robinson et al., 2019; Ortiz et al., 2018).

In this work, we present a novel neural network architecture to increase robustness to distributional changes (See Figure 1). Our framework combines conditional computation (Dumoulin et al., 2018; 2016; De Vries et al., 2017; Perez et al., 2018) with a task specific neural architecture for better domain shift generalization.

One key feature of this architecture is the ability to exploit extra information, often available but seldom used by current models, through a conditioning network. This results in models with better generalization, better performance in both independent and identically distributed (i.i.d.) and non- i.i.d. settings, and in some cases faster convergence. We demonstrate these methodological innovations on an aerial building segmentation task, where test images are from different geographic areas than the ones seen during training (Maggiori et al., 2017) and on the task of Tumor Infiltrating Lymphocytes (TIL) classification (Saltz et al., 2018).

We summarize our main contributions as follows:

- We propose a novel architecture to effectively incorporate conditioning information, such as metadata.

- We show empirically that our conditional network improves performance in the task of semantic segmentation and image classification.

- We study how conditional networks improve generalization in the presence of distributional shift.

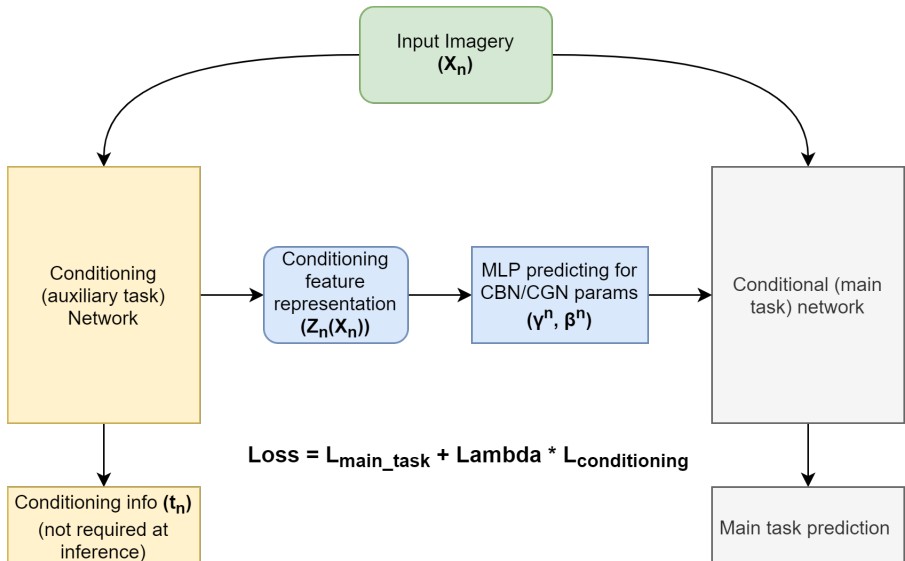

Figure 1: Conditional Networks

## 2 BACKGROUND AND RELATED WORK

**Self-supervised learning.** Self-supervised learning extracts and uses available relevant context and embedded metadata as supervisory signals. It is a representation learning approach that exploits a variety of labels that come with the data for free. To leverage large amounts of unlabeled data, it is possible to set the learning objectives such that supervision is generated from the data itself. The self-supervised task, also known as pretext task, guides us to a supervised loss function (Gidaris et al., 2018; Oord et al., 2018; He et al., 2019; Chen et al., 2020). However, in self-supervised learning we usually do not emphasize performance on this auxiliary task. Rather we focus on the learned intermediate representation with the expectation that this representation can carry good semantic or structural meanings and can be beneficial to a variety of practical downstream tasks. Conditional networks can be seen as a self-supervision approach in which the pretext task is jointly learned with the downstream task.

Our proposed modulation of a network architecture based on an auxiliary network's intermediate representation can also be seen as an instance of *knowledge transfer* (Hinton et al., 2015; Urban et al., 2016; Buciluǎ et al., 2006). Because the auxiliary network has an additional task signal – metadata prediction – information about this task can be transferred to the main task network.

**Conditional Computation.** Ioffe and Szegedy designed Batch Normalization (BN) as a technique to accelerate the training of deep neural networks (Ioffe & Szegedy, 2015). BN normalizes a given mini-batch $B = \{F_{n,.,.,.}\}_{n=1}^{N}$ of $N$ feature maps $F_{n,.,.,.}$ as described by the following Equation:

$$BN(F_{n,c,h,w}|\gamma_c, \beta_c) = \gamma_c \frac{F_{n,c,h,w} - \mathbb{E}_B[F_{.,c,.,.}]}{\sqrt{\text{Var}_B[F_{.,c,.,.}] + \epsilon}} + \beta_c, \qquad (1)$$

where $c, h$ and $w$ are indexing the channel, height and width axis, respectively, $\gamma_c$ and $\beta_c$ are trainable scale and shift parameters, introduced to keep the representational power of the original network, and $\epsilon$ is a constant factor for numerical stability. For convolutional layers the mean and variance are computed over both the batch and spatial dimensions, implying that each location in the feature map is normalized in the same way.

De Vries et al. (2017); Perez et al. (2018) introduced Conditional Batch Normalization (CBN) as a method for language-vision tasks. Instead of setting $\gamma_c$ and $\beta_c$ in Equation 1 directly, CBN defines them as learned functions $\beta_{n,c} = \beta_c(q_n)$ and $\gamma_{n,c} = \gamma_c(q_n)$ of a conditioning input $q_n$. Note that this results in a different scale and shift for each sample in a mini-batch. Scale ($\gamma_{n,c}$) and shift ($\beta_{n,c}$) parameters for each convolutional feature are generated and applied to each feature via an affine transformation. Feature-wise transformations frequently have enough capacity to model complex

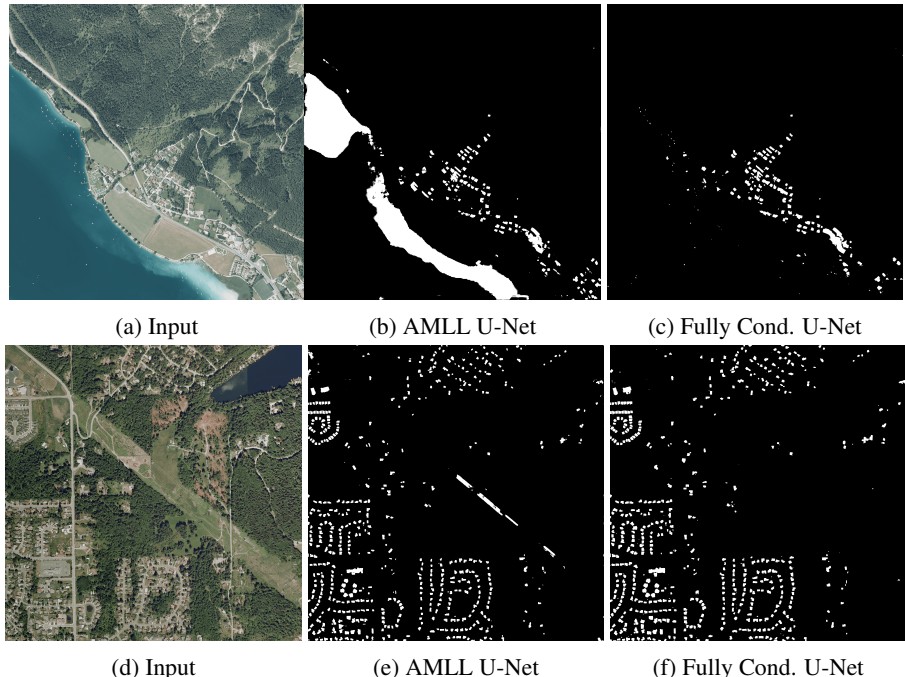

Figure 2: Qualitative Results. (a) Input from East Tyrol city, (b) AMLL U-Net segmentation results for (a), (c) Fully Cond. U-Net CGN segmentation results for (a), (d) Input from Bellingham city, (e) AMLL U-Net segmentation results for (d), (f) Fully Cond. U-Net CGN segmentation results for (d).

phenomena in various settings (Dumoulin et al., 2018). For instance, they have been successfully applied to neural style transfer (Dumoulin et al., 2016) and visual question answering (Perez et al., 2018). This kind of conditional computation scheme is not tied to the type of normalization used. Wu & He (2018) recently proposed Group Normalization (GN). GN divides feature maps into groups and normalizes the features within each group. GN only uses the layer dimension, hence its computation is independent of batch size. Ortiz et al. (2020) proposed Local Context Normalization (LCN) to encourage local contrast enhancement by normalizing features based on a spatial window around it and the filters in its group. Recently, Michalski et al. (2019) showed that Conditional Group Normalization (CGN) offer performance similar to CBN. In this work, we show results using CBN and CGN. Conditional normalization methods have been applied to tasks related to generalization, such as few-shot learning (Jiang et al., 2018; Tseng et al., 2020) and domain adaption (Su et al., 2020). Su et al. propose to use conditional normalization and an adversarial loss for domain adaption in object detection. In contrast to this work, we propose a method for implicit conditioning on an auxiliary task to leverage available metadata.

## 3 FORMULATION AND NETWORK ARCHITECTURE

### 3.1 PROBLEM ABSTRACTION

We first establish notation. Let $x$ be an image input, associated with a ground-truth target $y$ for the main task (e.g. a segmentation mask). Let available extra annotation for $x$ be denoted by $t$. The *main network* is trained to predict $y$ given $x$ and contextual information from an auxiliary network. The auxiliary network learns to predict $t$, also given $x$. Features $z$ of an intermediate layer of the auxiliary network are used to transform the main task network's layers using conditional normalization parameterized by a learned function of $z$.

The motivation for this method of implicit conditioning is the following:

1. Since $t$'s are only used as training targets, auxiliary annotation is not required at test time.
2. During training, the auxiliary network learns (via backpropagation) to *visually* capture information predictive of $t$. At test time, the auxiliary network reasons about the unavailable

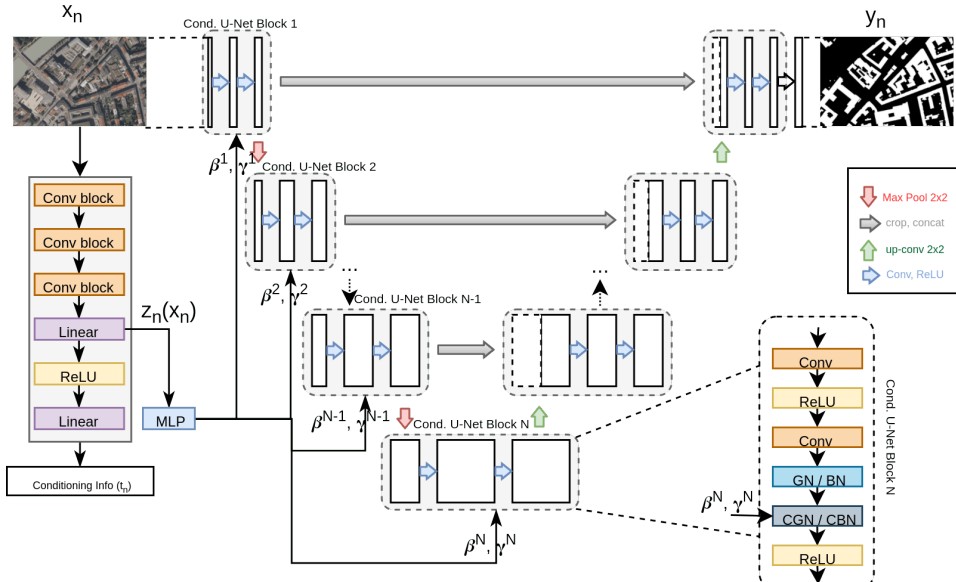

Figure 3: Conditional Network Architecture for U-Net

*t in terms of visual patterns that correlate with auxiliary annotations of training data.* Note that this allows the distribution of auxiliary information at test time to differ from the training data (see for example our experiments on out-of-distribution generalization in remote sensing in Section 4.1).

While the first statement is true for any multi-task architecture, the second statement describes the flexibility of the proposed method in leveraging auxiliary information of varying degrees of relevance. Obviously, the modulation will help most, if the auxiliary information is maximally relevant for the main task. Since the mapping from $z$ to the modulation parameters is trained with the main task's training signal, the network can learn to discard components of $z$ that are not useful for the main task. It is also possible for the network to learn a constant identity transformation of the main network's features in case no correlation is found. This reduces the potential of negative transfer learning between unrelated tasks common in multi-task learning (Ruder, 2017).

To provide an example of how this method can help to exploit inexpensive metadata. Consider the task of segmenting satellite imagery of different regions on the globe. We can use the prediction of geographic coordinates, which are often logged by default when building satellite imagery datasets, as the auxiliary task. In this case, the auxiliary network may learn to capture visual characteristics that are distinctive for each region in the training set, such as a predominance for smaller buildings. This would provide a useful inductive bias for the segmentation network, even for regions with very different coordinates. By using feature modulation to integrate this *contextual information*, we hypothesize that the main network can learn more general purpose features, which can be *attended to* based on the context.

## 3.2 NETWORK ARCHITECTURE

Our proposed architecture modification transforms any standard neural network with normalization layers into one that incorporates conditioning information $t$. In each convolutional block of the neural network we substitute the normalization layer by it's conditional counterpart. We refer to this family of networks as Conditional Networks. Figure 3 shows this extension applied to the popular U-Net Ronneberger et al. (2015) architecture. U-Net is an encoder-decoder network architecture with skip connections. Figure 3 shows the auxiliary network on the left modulating the modified U-Net on the right. The conditioning network is a convolutional architecture (LeCun et al., 1998) followed by a fully-connected layer that predicts metadata $t_n$ as a function of the input image $x_n$. The pre-activation features before the output layer are used as $z(x_n)$. The functions $\beta_{n,c}(z(x_n))$ and $\gamma_{n,c}(z(x_n))$ mapping $z(x_n)$ to the scale and shift parameters are implemented with a multi-

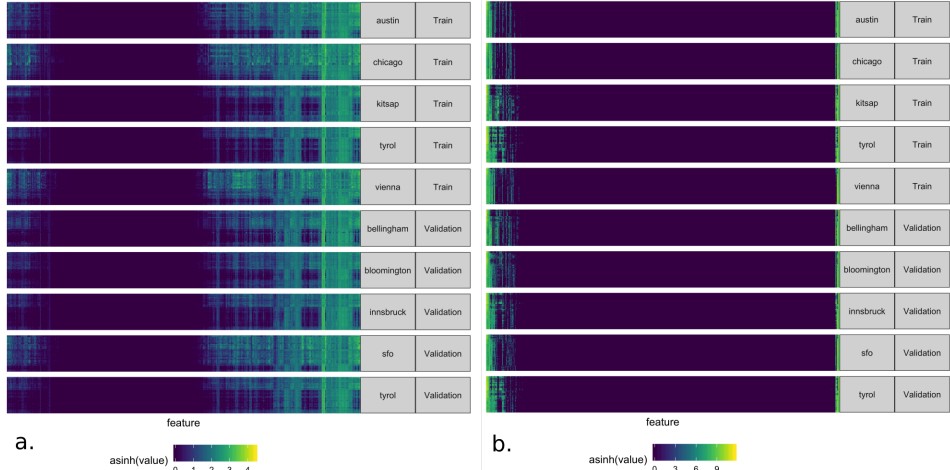

Figure 4: Heatmap of activations for (a) AMLL U-Net and (b) conditional U-Net. Cell $ij$ in row $i$ and column $j$ gives the activation of features $j$ on patch $i$. Rows and columns are sorted so that those that are more similar to one another appear side-by-side. Note that the color legends have different scales.

layer perceptron (MLP). Using the latent representations instead of directly using $t_n$ allows us to leverage combinations of features that were useful in localizing images from previously seen data, potentially improving generalization.

Because all its parts are differentiable, conditional networks can be trained end-to-end using gradient-based optimization. Our full objective is described in Equation 2, where $\alpha$ is a hyper-parameter balancing the main and auxiliary losses. $L_{\mathrm{main\_task}}$ represents a standard main task loss. $L_{\mathrm{main\_task}}$ depends on the task, such as Jaccard, cross-entropy, and dice for semantic segmentation. $L_{\mathrm{conditioning}}$ ensures the conditioning networks correctly predicts $t_n$.

$$L_{\mathrm{cond.\_net}} = L_{\mathrm{main\_task}} + \alpha \cdot L_{\mathrm{conditioning}} \qquad (2)$$

## 4 EXPERIMENTS

We study the following hypotheses:

**H1: Generalization through context.** Explicit incorporation of conditioning information improves generalization in semantic segmentation and image classification tasks.

**H2: Interpretability.** The features learned by the conditioning and the main task network reflect context-specific and context-invariant information, respectively.

### 4.1 CONDITIONAL NETWORKS FOR SEMANTIC SEGMENTATION OF AERIAL IMAGES

To study hypotheses H1 and H2 we focus on the Inria Aerial Image Labeling Dataset. This dataset was introduced to test out-of-distribution generalization of remote-sensing segmentation models (Maggiori et al., 2017). It includes imagery from 10 dissimilar urban areas in North America and Europe. Instead of splitting adjacent portions of the same images into training and test sets, the splitting was done city wise. All tiles of five cities were included in the training set and the remaining cities are used as test set. The test set also has variation in illumination, landscape, and time, making it well-suited to evaluate out-of-distribution generalization. The provided imagery comes orthorectified and has a spatial resolution of 0.3m per pixel covering $810km^2$ ($405km^2$ for training and $405km^2$ for the test set) on an evenly spaced grid. Images were labeled for the semantic classes of *building* and *not building*. We use geographical coordinates of the images as target data for the auxiliary network (see Section 3).

Table 1: Inria Test Set Performance.

| Method | Bellingham | | Bloomington | | Innsbruck | | San Francisco | | East Tyrol | | Overall | |
|---|---|---|---|---|---|---|---|---|---|---|---|---|
| | IoU | Acc. | IoU | Acc. | IoU | Acc. | IoU | Acc. | IoU | Acc. | IoU | Acc. |
| AMLL U-Net | 65.37 | **96.53** | 55.07 | 95.83 | 67.62 | 96.08 | 72.80 | 91.00 | 67.00 | 96.91 | 67.98 | 95.27 |
| U-Net + GN | 55.48 | 93.38 | 55.47 | 94.41 | 58.93 | 93.77 | 72.12 | 89.56 | 62.27 | 95.73 | 63.71 | 93.45 |
| U-Net + LCN | 63.61 | 96.26 | 60.47 | 96.22 | 68.99 | 96.28 | **75.01** | **91.46** | 68.90 | 97.19 | 69.90 | **95.48** |
| Cond. U-Net CBN | 60.38 | 95.91 | 48.84 | 95.20 | 65.99 | 95.97 | 72.74 | 90.88 | 69.70 | 97.24 | 66.54 | 95.04 |
| Cond. U-Net CGN | 63.10 | 96.15 | 56.43 | 95.84 | 68.70 | 96.23 | 74.31 | 91.28 | 66.71 | 97.03 | 68.66 | 95.31 |
| Fully Cond. U-Net CBN | 65.91 | 96.49 | 57.24 | 95.95 | 63.14 | 95.71 | 73.89 | 90.83 | 69.24 | 97.21 | 68.50 | 95.24 |
| Fully Cond. U-Net CGN | **66.98** | 96.52 | **63.27** | **96.33** | **69.80** | **96.24** | 73.88 | 90.75 | **70.77** | **97.35** | **70.55** | 95.44 |
| $nont_n$ Cond. U-Net | 62.97 | 96.10 | 53.08 | 95.03 | 65.61 | 95.73 | 71.87 | 90.32 | 65.66 | 97.75 | 66.33 | 94.78 |

Table 2: Distribution Shift Generalization Gap.

| Method | Val. set | Test set | Gen. Gap |
|---|---|---|---|
| | IoU (%) | IoU (%) | IoU (%) |
| AMLL U-Net | 71.87 | 67.98 | 3.89 |
| U-Net + GN | 71.38 | 63.71 | 7.67 |
| Cond. U-Net CBN | 72.15 | 68.50 | 3.65 |
| Cond. U-Net CGN | **72.77** | **70.55** | **2.22** |

For H1, we compare model performances using the standard benchmark training-test split. For H2, we perform an exploratory visualization of the feature-activation maps for the different models.

**Generalization via conditioning.** We used the Inria standard train-test split to see whether conditioning information helps out-of-distribution generalization. From the training set we reserved five images of each city for the validation set as suggested by Maggiori et al. (2017).

For this set of experiments we trained our conditional U-Net presented in Figure 3 end-to-end from scratch. We used as segmentation network the AMLL U-Net as described by Huang et al. (2018), which is a version of U-Net with fewer filters. The AMLL U-Net was the winning entry and top of the Inria leaderboard and we use it as baseline for comparison in this section.

The Conditional U-Net was trained exactly as AMLL U-Net, but without any data augmentation technique. We used the standardized latitude and longitude of the center pixel of each patch as the conditioning information to be predicted by the conditioning network and the mean-squared error (MSE) as the conditioning loss $L_{\text{conditioning}}$ and cross-entropy as segmentation loss $L_{\text{main\_task}}$. All training details and a figure showing histograms of final Intersection-over-Union (IoU) scores of the different models can be found in the Appendix sections A.1 and A.2.

Table 1 shows the performance of the AMLL U-Net baselines and different variations of our proposed architecture on the test set. Conditioning uniformly improved segmentation performance over the corresponding conditioning-free models. This empirically validates our hypothesis H1 about generalization through context. We identify as *Cond. U-Net* those models in which both the encoder and decoder are modulated, which yielded a small gain in performance over just modulating the encoder or decoder alone. We recommend conditioning all blocks since the extra computational cost is very small. Modulating using CGN consistently outperforms CBN.

$t_n$ **Matters.** An experiment using the conditioning network without the auxiliary task of predicting $t_n$ shows a deterioration of performance relative to the baseline U-Net as shown in Table 1. We see this as evidence for the importance of guiding the learning process of the latent conditioning representation $z(x_n)$.

Table 2 shows the generalization gap as the difference between the validation set (i.i.d.) and the test set performance. Notice how the models' performances consistently degrade when we evaluate them on cities not seen during training. Conditioning substantially reduces the generalization gap induced by the distribution-shift between the training and test sets, yielding evidence for hypothesis H1.

Figure 2 shows a qualitative comparison in the performance of the proposed network. The baseline labels a beach (2a) and power lines (2d) as buildings while the Conditional U-Net does not.

## 4.2 INTERPRETATION OF CONDITIONING FEATURES.

To evaluate hypothesis H2, we analyze patterns of activations across these experiments. This hypothesis is interesting for several reasons,

- It serves as a sanity check for the proposed architecture, ensuring that supervision from conditioning information leads to features that do indeed distinguish between cities.
- It sheds light on the potential of using conditioning information to facilitate learning of generalizable features by intentionally learning context-dependent features.
- We can begin to characterize how generalization occurs, by identifying which training cities a test patch "looks" like.

As our first approach towards characterizing feature context-dependence, we associate activations with underlying conditioning information. We apply the following procedure for a few models,

- Compute all activations at a pre-specified layer in the network for all patches within an epoch.
- Compute the norm of activations for each feature map in that layer.
- Arrange these values into a patch $\times$ feature matrix, and visualize using a heatmap.

For U-Net models, we focus on the "bottom" of the U, which has a large number of filters with small spatial extent. In the conditional U-Net, we additionally compute the activations from the last layer before prediction of patch coordinates[1] If we notice separation of patch activations according to conditioning information, we deduce that the learned feature maps are not invariant to that conditioning context.

The learned activations are displayed in Figure 4 [2]. We find, that individual features that activate for a patch in one city tend to activate in a large fraction of patches across all cities. Further, across similar cities, patterns of activation are similar, for both conditioned and unconditioned models. Consider relative similarity between Chicago and San Francisco, for example [3].

For the conditioned model, the large majority of features are zero, across all patches. However, when they are nonzero, their values tend to be larger than the typical activation in unconditioned models. From these observations, we conclude that the improved generalization ability of the conditional U-Net is not due to any ability to learn features that are more invariant to the identity of the corresponding city. Instead, it appears that the conditional U-Net learns a smaller collection of features that are ultimately more useful in the downstream segmentation task. We speculate that having fewer active features for any prediction allows for sharper predictions, preventing "blurring" that could result from averaging across feature maps. More details around H2 are presented in the analysis of feature variance and Figure 10 in the Appendix section A.3. In summary, Figure 4 gives evidence against H2, suggesting that conditional network architectures do not neatly segregate context-specific and context-dependent features. Figure 4 and 10 both suggest that the learned features are qualitatively different between the architectures.

## 4.3 CONDITIONAL NETWORKS FOR TUMOR INFILTRATING LYMPHOCYTES CLASSIFICATION

We also test Conditional Networks for the task of tumor infiltrating lymphocytes (TIL) Classification. During the cancer diagnosis and treatment process, a patient may have a biopsy, which produces a diagnostic tissue sample. Using this sample, a slide is prepared and examined under a microscope by a pathologist to understand both how to treat the disease and to provide a prognosis for the patient's future. Virtually all cancer patients undergo these biopsies, producing large volumes of these pathology slides.

A significant feature in these images is tumor infiltrating lymphocytes (TILs), which are types of immune cells that move into a tumor to try to attack the cancer. The quantification of TILs is

---

[1]Since these features have no spatial extent, we do not take any norms.

[2]Tyrol/train refers to West Tyrol city, Tyrol/Validation refers to East Tyrol city as established in the Inria dataset

[3]We also present the associated t-SNE projection in section A.3

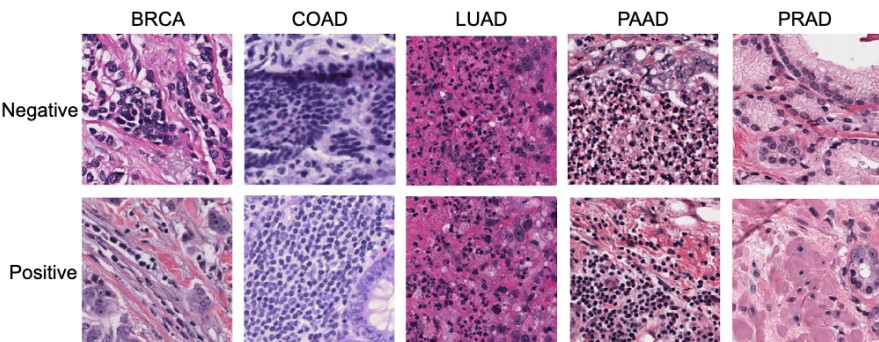

Figure 5: Example of negative and positive TIL patches for different cancer types. The desired task is to properly classify a patch as TIL negative or TIL positive independently of the cancer type.

well known to have prognostic value in many contexts (Fridman et al., 2012; Angell & Galon, 2013) because understanding patient immune response to tumors is becoming increasingly important with the growth of cancer immunotherapy. Features such as TILs can be quantified through image analysis and deep learning algorithms (Saltz et al., 2018; Klauschen et al., 2018). In Saltz et al. (2018), a convolutional neural network (CNN) architecture is systematically optimized to carry out classification of nuclei from pathology images. This led to the release of a dataset consisting of TIL maps corresponding to roughly 5,000 whole slide images from The Cancer Genome Atlas (TCGA). Individual slide images were split into $100 \times 100$ patches and the tasks is to classify TIL patches as positive or negative in the exact same dataset setup as Saltz et al. (2018). The training set consists of 86,154 patches that were manually annotated with TIL classification (Saltz et al. (2018)). There are 64,381 TIL negative patches and 21,773 TIL positive patches. All patches are in $100 \times 100$ pixel resolution, 20 times magnification, and are annotated as TIL positive or TIL negative. Examples of the images and their labels are given in Figure 5.

These training images represent seven different cancer types: invasive carcinoma of the breast (BRCA), colon adenocarcinoma (COAD), lung adenocarcinoma (LUAD), pancreatic adenocarcinoma (PAAD), prostate adenocarcinoma (PRAD), skin cutaneous melanoma (SKCM), and endometrial carcinoma of the uterine corpua (UCEC). The cancer type is the conditioning informations or metadata ($t_n$) used to train the conditioning network. We use another 652 patches as our validation set, and 900 manually annotated patches from twelve cancer types in total as the testing set. The twelve cancer types are the seven listed above, as well as five novel ones never seen during training (urothelial carcinoma of the bladder (BLCA), cervical squamous cell carcinoma and endocervical adenocarcinoma (CESC), lung squamous cell carcinoma (LUSC), rectal adenocarcinoma (READ), and stomach adenocarcinoma (STAD)).

**Methods tested.** We train the VGG 16-layers network as baseline (Simonyan & Zisserman (2014)). VGG networks have been shown to work well for pathology image classification (Xu et al. (2015); Hou et al. (2016)). We then build the conditional version of VGG16 which we refer to as Conditional VGG16. Conditional VGG16 is created in a similar way as conditional U-Net and share a similar conditioning network. In this experiments the conditioning network is trained to perform image classification for the type of cancer ($t_n$) using a seven-dimensional softmax layer. As with conditional U-Net, we predict $\beta_{n,c}(z(x_n))$ and $\gamma_{n,c}(z(x_n))$ from features in the conditioning network. $\beta_{n,c}(z(x_n))$ and $\gamma_{n,c}(z(x_n))$ are then used to modulate VGG16 architecture. We use binary cross-entropy as $L_{\text{main\_task}}$ and multiclass cross-entropy as $L_{\text{conditioning}}$. Training details are provided in the supplemental material.

**Results.** Table 3 shows the results for the Tumor infiltrating Lymphocyte classification task using different approaches. Augmenting VGG16 using conditional networks improves its performance by a large margin allowing us to obtain state-of-the-art performance in the task, even improving over previous top performing approaches. It is also important to remember that $t_n$ (cancer type) is only required during training. Inference only requires the input image, as in competing methods.

To show the importance of the conditioning network we trained a version of conditional VGG16 where conditioning parameters are predicted directly from the cancer type softmax prediction from

Table 3: TIL Classification Test (generalization set) Results

| Model | AUC (%) |
|---|---|
| Spatial (Saltz et al. (2018); Hou et al. (2019)) | 86.16 |
| Inception (Szegedy et al. (2016); Patton et al. (2019)) | 89.9 |
| MENNDL (Patton et al. (2019)) | 86.16 |
| **Conditional VGG16 (ours)** | **92.91** |
| Direct Cond. VGG16 | 50.16 |
| Cond. VGG16 alpha 0 | 77.18 |

the conditioning/auxiliary network and we refer to it as "direct cond. VGG16" on Table 3. As expected, doing so does not work well. Properly predicting for the conditioning task is also very important for conditional networks to work well. When we only optimize for $L_{\mathrm{main\_task}}$ performance degrades as it is shown on Table 3 under "Cond. VGG16 alpha 0".

## 5 DISCUSSION AND CONCLUSIONS

We have presented *conditional networks*, a family of neural networks that leverages conditional information for improved performance and better generalization. Conditional networks can be applied to any network architecture that uses normalization layers. We showed how the performance of two widely adopted network architectures (U-Net and VGG16) can be greatly improved by applying conditional networks for both semantic segmentation and image classification. We have shown that conditional networks consistently reduce the generalization gap observed when there is a shift of the underlying distribution at test time. After carefully studying the network feature activations, we found that the improved generalization ability of the proposed network is not due to the ability of learning more invariant features. It appears, instead, that the conditional network learns a smaller collection of features more relevant to the task. Conditional networks exploits available extra annotation $t_n$ while training, however $t_n$ is not required during inference. It is not always obvious which choice of $t_n$ helps the most to better estimate $z_n$. Future work involves studying how the choice of $t_n$ influences performance in a larger set of tasks.

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

## A    APPENDIX

### A.1    IMPLEMENTATION DETAILS

**Training Details Aerial Image labeling Experiments and Conditional U-Net**    We trained all networks for the task of aerial image labeling using 572x572 randomly sampled patches from all training image tiles. We used the Adam optimizer (Kingma & Ba, 2014) with a batch size of 12. All networks were trained from scratch with a learning rate of 0.001. Every network was trained for 100 epochs. We keep the same learning rate for the first 60 epochs and decay the rate to 0.0001 over the next 40 epochs. In every epoch 8,000 patches are seen. Binary Cross-Entropy was used as the segmentation loss function. For the conditioning part of the conditional network we used the latitude and longitude of every patch center pixel as $t$. Latitude and longitude were standardised to be from -1 to +1. The dimension of $z$ is 4374 (9x9x54). $z$ is the input of the MLP predicting for $\gamma$ and $\beta$. We used 1024 hidden units in the MLP.

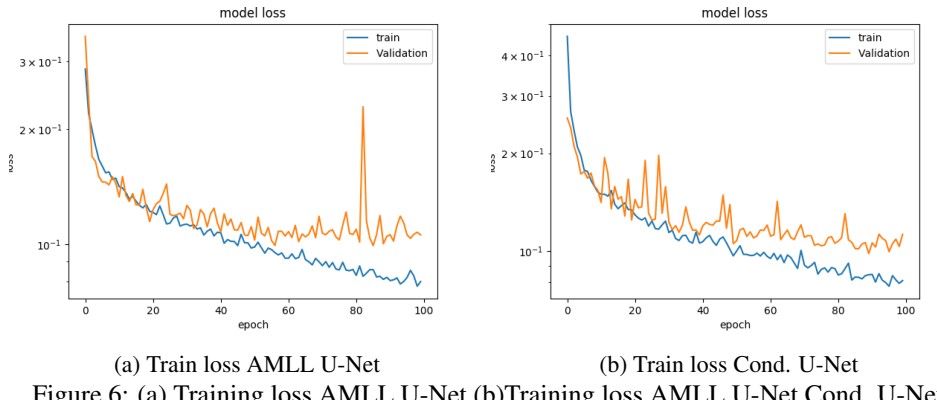

(a) Train loss AMLL U-Net                        (b) Train loss Cond. U-Net

Figure 6: (a) Training loss AMLL U-Net (b)Training loss AMLL U-Net Cond. U-Net

Figure 6 shows the learning curves of the baseline U-Net and the fully conditional U-Net (CGN). The AMLL U-Net seems more prone to overfitting than our conditional U-Net.

**Training Details Conditional VGG16.**    We trained all training $100 \times 100$ TIL patches from the training set. We used the Adam optimizer (Kingma & Ba, 2014) with a batch size of 16. All networks were trained from scratch for 20 epochs with a learning rate of 0.001. Binary Cross-Entropy loss was used for the $L_{main\_task}$ (TIL classification) and multiclass cross-entropy for the $L_{conditioning\_task}$ (cancer type classification) loss function. For the conditioning part of the conditional network we used the cancer type as metadata $t$. $\alpha$ for the conditional loss was 0.8. No data augmentation was performed.

### A.2    DETAILED PERFORMANCE CONDITIONAL U-NET

Figure 7 shows the overall and per-city performance of the models. The fully-conditional U-Net variant using CGN consistently outperforms the other models on every city in the test set. The CGN variant where only the encoder is conditioned has a stronger overall performance and generalizes significantly better than the conditional U-Net variants using BN. This is consistent with the observation of Wu & He (2018) that (regular) GN outperforms BN in segmentation tasks.

### A.3    NETWORK FEATURE ACTIVATIONS

Figures 8 and 9 are the t-SNE figures obtained from inspecting the activations of the bottom of the U in the U-Net for both the AMLL U-Net and Conditional U-Net CGN. Figure 8 shows the t-SNE associated to sample images from the same cities in the training set [4]. Figure 8 shows the t-SNE associated to sample images from both training and test set [5].

---

[4]Tyrol legend refers to West Tyrol as it refers to the training set

[5]Tyrol+Train refers to West Tyrol and Tyrol when train is false refers to East Tyrol

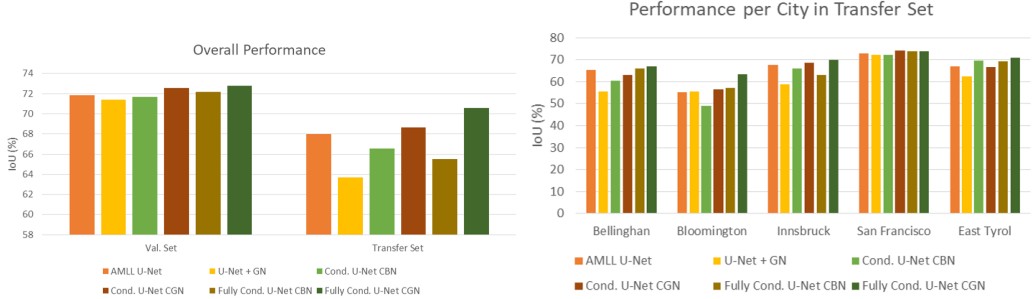

(a) Histogram of overall performance

(b) Histogram of per city performance

Figure 7: (a) Histogram of overall performance for both validation and test set (b) Histogram of per city performance

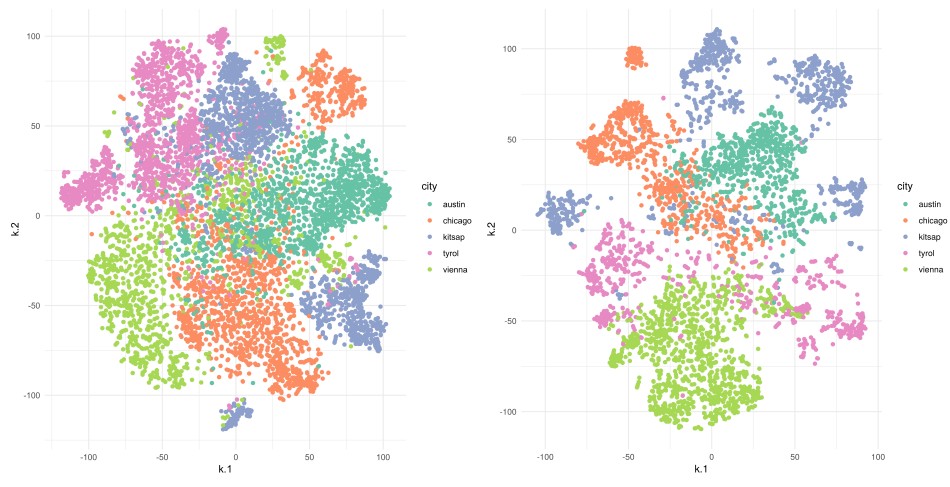

(a) t-SNE AMLL U-Net train set

(b) t-SNE Cond. U-Net train set

Figure 8: t-SNE based on activations obtained using image tiles from the cities in the training set. (a) AMLL U-Net train set (b) Cond. U-Net CGN train set

In light of the differences in scale and sparsity of feature activations in the conditional and AMLL U-Nets in Figure 4, we zoom into individual features of interest in Figure 10 [6]. It is not surprising that the overall $y$-axis scale differs between the two networks – this is clear from the legend of Figure 4. However, for both rows of Figure 10, we observe that there are more instances of high variation in activation across patches in the conditional U-Net, compared to the AMLL U-Net. In the conditional U-Net, there seem to be features that, while typically active, are often exactly or near zero, and similarly, features that are typically inactive, but occasionally spike. This type of variation is even observed within individual cities. This suggests a type of specificity in the learned features. Rather than activating slightly more or slightly less across all patches, features seem sensitive to particular features within the patches that they activate. While conditional U-Net features are not invariant to conditioning data, they do appear to be more specialized.

---

[6]We refer to this figure while discussing hypothesis 2 in the section 4.2

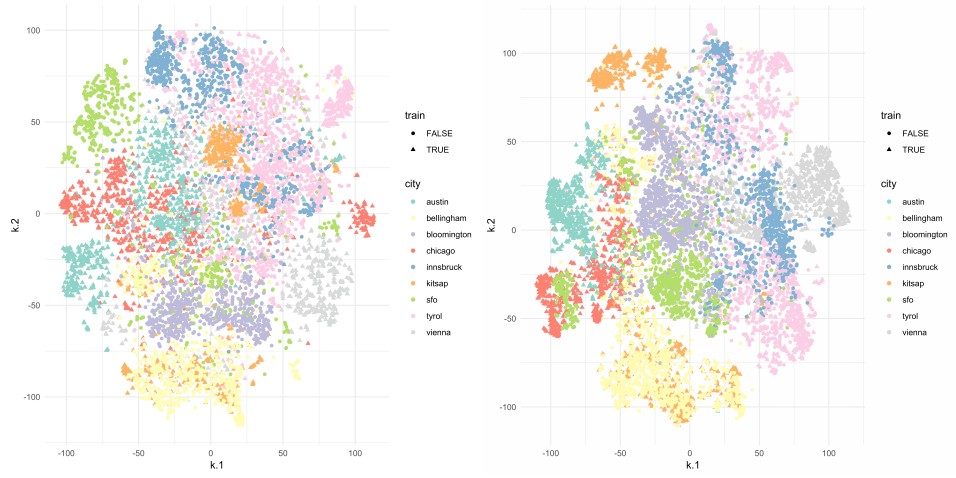

(a) t-SNE AMLL U-Net train and test sets    (b) t-SNE Cond. U-Net train and test sets

Figure 9: t-SNE based on activations obtained using image tiles from cities in both training and test sets. (a) AMLL U-Net train and test sets (b) Cond. U-Net CGN train and test sets

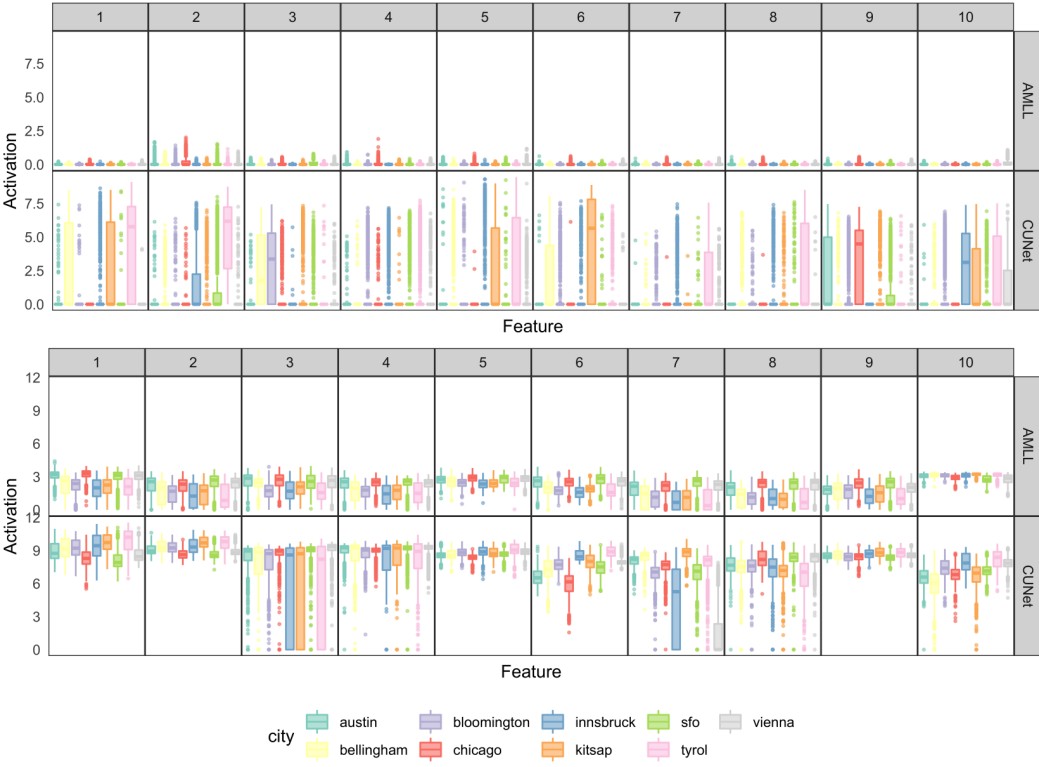

Figure 10: The (top) least and (bottom) most variable features, according to interquartile range, in the AMLL and conditional U-Nets, after filtering to those features that are active in at least 10% of patches. Columns 1-10 give the 10 most and least variable features, for least and most variable features, respectively. The $y$-axis is the activation for each feature. Patches are split into individual cities, and a boxplot of each city's activation values is given.

