# OpenReview forum: "Conditional Networks"
_ICLR.cc/2021/Conference — Reject_

### Official Review · AnonReviewer4 · 2020-10-28
**Review Comment**

**Rating:** 3
**Confidence:** 5

**Review:**

- Summary:

This paper aims to tackle the out-of-distribution generalization problem where a model needs to generalize to new distributions at test time. The authors propose to utilize some extra information like the additional annotations as the conditional input and output the affine transformation parameters for the batch normalization stage. This extra information helps the backbone network get a more general representation from the training set thus the model is robust to the distribution shift when testing. Experiments are conducted on the Aerial Image Labeling and the Tumor-Infiltrating Lymphocytes datasets which correspond to the image segmentation and classification task respectively.

Comments:
1. The Conditional Batch Normalization (CBN) mechanism has been used in some generalization relevant tasks like domain adaptation detection [1] and few-shot learning [2]. The paper misses the insightful ideas, and only simply combines the CBN into different backbones and lacks theoretical analysis or empirical evidence about how the additional information and the conditional network can help to improve the model’s generalization ability. I think more ablation studies to demonstrate the effectiveness of the proposed conditional network and choices of the “metadata” or additional information in a dataset are needed.
[1] Adapting Object Detectors with Conditional Domain Normalization
[2] Cross-Domain Few-Shot Classification via Learned Feature-Wise Transformation

2. The paper demonstrates the experiments on two tasks (i.e. semantic segmentation, image classification), but the proposed conditional network is different and does not have a unified architecture. So the concept of the conditional networks is over-packaged.

3. The motivation proposed framework is not clear. The conditional learning assumption is a straight-forward idea for machine learning or deep learning, so what is the main novel mechanism proposed in the paper is not clear.

Questions:

1. Is there now an existing standard benchmark of the out-of-distribution (OOD) generalization task? I noticed that this paper does not compare with other OOD generalization methods.

2. What is the meaning of Figure 4? It seems that the explanation is not clear.

3. From this paper, the additional information of the Inria Aerial Image Labeling dataset is the geographical coordinates. But what is the additional information or “metadata” of the Tumor-Infiltrating Lymphocytes dataset? I think it is an important issue but I did not find an answer from this submission.

---

> ### Author Response · Authors · 2020-11-23
> **Clarifications and added related work discussion**
>
> We appreciate your comments and insightful suggestions.\
>  To address your concerns:
> 1. **The concept of the conditional networks is over-packaged:** We consider the fact that our proposed approach works for different tasks (classification and segmentation) and using different conditioning networks an advantage. Rather than proposing a specific architecture, conditional networks are a self-supervised framework that’s not tied to specific architectures.
> 2. **What’s the main novel mechanism proposed?**
> The auxiliary network that generates the modulation parameters is trained on an additional prediction task, while it receives the **same input modality** as the main network. Besides not requiring the additional modality at test time, we believe that the network can learn visual characteristics that correlate with the auxiliary modality. This can yield a useful prior for the main task. How helpful this prior is, is obviously dependent on the choice of auxiliary data. We tried to further clarify our contribution and the motivation behind it in the revised manuscript.
> 3.**Is there now an existing standard benchmark of the out-of-distribution (OOD) generalization task?** We are not aware of a go to OOD benchmark. We would also like to acknowledge that conditional networks are constrained to the availability of metadata relevant to the task. Hence, for certain OOD tasks conditional networks won't be applicable. Note, that many works on OOD generalization are concerned with synthetic benchmarks that have very precise definitions of their OOD properties. Our focus in the present manuscript is to address the less clearly defined OOD nature of real-world applications.
> 4. **Meaning of Figure 4:** We have extended the discussion of Figure 4, clarifying its meaning and relationship with Hypothesis 2. New section 4.2 provides a deeper analysis on hypothesis 2.
> 5. **What’s “metadata” of the Tumor-Infiltrating Lymphocytes dataset?** As stated under paragraph heading “Methods tested.”  In Section 4.2, the metadata used for the TIL classification experiment is the cancer type the patient is suffering from. We have added details clarifying this point in section 4.2. The conditioning task (tn) performed by the conditioning network is to classify the cancer type using a 7-dimensional softmax output layer (1 out of 7 types occurring in the training set).
> 6. **CBN mechanism has been used in some generalization relevant tasks.** Thanks for sharing these related works. We have added a brief discussion on this in the related work section. The proposed idea in [1] of leveraging an adversarial loss and conditional domain normalization could potentially be combined with our proposed framework for leveraging auxiliary data/modalities.
>
> **References**
> 1. Adapting Object Detectors with Conditional Domain Normalization

---

### Official Review · AnonReviewer3 · 2020-10-28
**Powerful approach to modulate CNN feature map activations by means of an auxiliary target to increase OOD generalisation with some clarifications required in the experimental evaluation**

**Rating:** 6
**Confidence:** 4

**Review:**

### Summary:
This submission proposes an approach to modulate activations of general convolutional neural networks by means of an auxiliary network trained on additional metadata to a dataset. The specific goal is to improve out-of-distribution (OOD) generalisation. This *conditional network* approach is illustrated for two standard convolutional neural network (CNN) architectures, U-Net and VGG, on two benchmark datasets suitable for OOD detection, the Inria Aerial Image Labeling Dataset and the Tumor Infiltrating Lymphocytes classification dataset. The conditional network approach yields favourable results compared to competing segmentation as well as classification networks and exhibits a reduction of the generalisation gap compared to the baseline methods.

### Strengths:
- Significance / Novelty: The conditional network uses a latent / intermediate representation *z* of an auxiliary network solving an auxiliary task *T* (as usually considered in the context of self-supervised learning) to learn parameters used to perform an affine transformation of feature map activations in the main neural network solving the main task *Y*. The approach combines self-supervised learning with conditional normalisation of activations. As far as I can judge, the proposed idea is original and novel.
- A particular benefit of the proposed approach is that, at test time, only the latent representation *z* is required (provide by the trained auxiliary network) without the necessity of providing metadata *t*.
- Relevance / Potential impact: The presented idea seems to improve performance and training of standard CNN architectures and can be used in general CNN architectures, which might render the approach applicable in a great variety of segmentation / classification tasks and relevant for a wider audience.
- Clarity: In my opinion, this paper is very well-written, self-contained and offers a good description of the approach and adequate experimental evaluation of the made claims with some exceptions outlined below.


### Weaknesses:
- Technical quality:  My major concern with respect to the evaluation is the comparison to the baseline. In the paper by Huang et al. on AMLL U-Net higher IoU scores and accuracies on the transfer (test) set are reported (table 2, page 5 in their paper) compared to table 1, which in fact exceed the performance results of the proposed method. This raises the question whether the baseline methods are properly trained. From the description in the main text and appendix (section A.1), the only difference w.r.t. Huang et al. consists in using data augmentation. Is there a particular reason why the training deviates from the settings outlined in Huang et al.?
- In connection to the previous point, also no details on hyperparameter tuning are provided and all networks were trained with the same settings. However, the differences in the methods might require different hyperparameter settings to provide the best performance. Were other hyperparameters considered in the outlined experiments?
- Using the geocoordinates as metadata *t_n* does not strike me as a particularly useful choice, as it appears unlikely that a representation of “location” is learned but rather some random features are extracted which are known to be beneficial, too (as shown e.g. by Rahimi and Recht). In my opinion, relative closeness in geocoordinates should have little information on input aerial regions. The second example with *t_n* being the cancer type is much more convincing.

- Clarity: Figure 2 and 4 are not explicitly addressed, put into context with other results or discussed in the main text (fig. 4 mentioned in the appendix). After carefully studying the paper, one might make a connection between the results in table 1 and figure 2. But although paragraph *”Interpretation of conditioning features”* (p. 7) seems to relate to figure 4, the results are difficult to interpret without any further discussion or comments on it. Could the authors elaborate more on Hypothesis 2 and Figure 4? I believe a more explicit / clearer discussion would improve the quality of the paper quite a bit.
- In connection to the previous point, the reasoning for the following conclusion (p. 8) should be made more clearer (and maybe connected to section A.3 and figure 10 in the appendix): *“After carefully studying the network feature activations, we found that the improved generalization ability of the proposed network is not due to the ability of learning more invariant features. It appears, instead, that the conditional network learns a smaller collection of features more relevant to the task.”*
- I might have missed it, but I am not sure what the difference between *“Cond. U-Net”* and *“Fully Cond. U-Net”* is. As the former already modulates all blocks in the encoder and decoder, what does *“fully cond.”* implicate?


### Additional Feedback:
- Figure 2 caption: *“(e) Fully Cond. U-Net CGN”* should read *“(e) AMLL U-Net”*.
- Figure 4: I would suggest using the same scale range for both plots, to make the difference more prominent. This change might be beneficial for the points mentioned above, too.
- Page 5, last sentences: *“[…] (see Section 4.1).”* This is a self-reference to the same section. Quite likely the reference was supposed to be *“Section 3.1”*.
- Tables 1 and 2, (overall) IoU score on transfer set for “U-Net + GN”: There is probably a small typo what concerns the values *“63.71”* (table 1) and *“63.79”* (table 2). If I understand correctly, these values should be exactly the same.
- Page 8, results paragraph, first sentence: *“[…] Tumor-infiltrating Lymphocyte […]”* while before *“[…] Tumor Infiltrating Lymphocyte […]”* was used (without the hyphen).

### Recommendation:
I enjoyed reading this submission and think that the idea of “conditional networks” constitutes a novel and relevant contribution to improve OOD generalisation. In particular, I believe the proposed approach might be impactful in a variety of related methods and applications as it can be used in general CNN architectures. However, there are some concerns and questions outlined above which I believe need to be addressed / adapted in order to accept this paper. My initial rating is weak accept, but I am willing to raise my rating if the authors can address these aspects.

### Post-Rebuttal:
I would like to thank the authors for addressing the questions and concerns. I still believe that the general idea of “conditional networks” might pose a relevant contribution to improving OOD generalisation. However, after rereading the submission, reading the other reviews, and taking the rebuttal into consideration, I think there are some aspects which need some revision and clarification. I comment on this in more detail below. Therefore, I stand with my initial rating of borderline, but I would like to encourage the authors to revise their paper taking the points raised by the reviewers into consideration and submit again.

- **Section 4.2, discussion of figure 4 and hypothesis 2:** I thank the authors for the added discussion. However, the results are a bit at odds with the premise of the proposed approach, in my opinion. The bullet points which detail why hypothesis 2 could be interesting, are refuted by the results presented below in that section. I believe the result that activations look different (what concerns scale of activation and which features are active) by itself is less surprising as the approach tackles the normalisation of activations explicitly. But more importantly, other than that, I would say the activation patterns for different cities look qualitatively similar in both models and does not align well with the story of the paper. So, in my opinion, the activation patterns on the left (AMLL U-Net) in figure 4 look very similar for all cities, and also the patterns on the right (Cond. U-Net) look quite alike for different cities. Therefore, the interpretation of these results in the context of conditioning on auxiliary information remains speculative. I believe this part of the submission requires a careful reconsideration.

- **Geocoordinates as metadata *t_n*:** A potential reason for the difficulties in the previous point could be the choice of metadata *t_n* in the segmentation example. I thank the authors for elaborating again on the choice. Still, I am not convinced that this is the most suitable choice to present the advantages of the proposed approach. If the conditioning network really just performs city ID classification, I would not be sure that any kind of useful (generalisable) features are extracted. This could be a potential explanation why no “conditioning influence” on the results in figure 4b is observed.

- **Difference “Cond. U-Net”:** I should’ve been more explicit in my question. On page 6, last paragraph of “Generalization via conditioning” it reads: *”We identify as Cond. U-Net those models in which both the encoder and decoder are modulated, which yielded a small gain in performance over just modulating the encoder or decoder alone.”* If I see correctly, “Cond. U-Net” should be replaced by “Fully Cond. U-Net”, as the provided answer suggests, too.


### References:
- Rahimi and Recht, “Random Features for Large-Scale Kernel Machines”, NeurIPS 2007.
- Huang et al., "Large-scale semantic classification: outcome of the first year of inria aerial image labeling benchmark", 2018.

---

> ### Author Response · Authors · 2020-11-23
> **Thank you for the detailed constructive feedback**
>
>
> We truly appreciate your thoughtful comments and the detailed feedback, which helped us in revising the manuscript. We would like to address your concerns:
>
> 1. **Difference in AMLL U-Net reported performance.**
> The original AMLL U-Net paper reports the performance after performing extensive data augmentation. However we do not perform any data augmentation since we aim to merely evaluate the aggregated value of conditional networks. We will make sure this is clarified in the document.
> 2. **All networks were trained with the same settings, were other hyperparameters considered in the outlined experiments?**
> All main task networks were trained with the same settings as the baseline network for fair comparison. We agree that doing more extensive hyper-parameter tuning could further improve conditional networks performance.
> 3. **Geocoordinates as metadata t_n.**
> We agree that geocoordinates are kind of weak in terms of additional supervision for the segmentation task, but the Inria dataset didn’t provide more metadata. Our intuition for why they might be useful, is that the network may discover visual characteristics (since the input is visual) of smaller regions within a city, since that would help to predict longitude and latitude. This in turn could yield a useful prior for the segmentation task. Think “this looks like neighborhood X in city Y from the training set, which had mostly smaller residential buildings with flat roofs.”. We tried to make that motivation more explicit in our revision.
> We raise a great point. Because each city in the training set is very far apart, in practice, we ended up seeing the conditioning network end up mostly predicting the city.  City ID classification is what probably occurs in the conditioning network and it might be a good idea to state it that way.
> 4. **Elaborate more on Hypothesis 2.**
> We have extended the discussion of Figure 4, clarifying its relationship with Hypothesis 2. Please *refer to new section 4.2* in the updated document.
> 5. **Difference between “Cond. U-Net” and “Fully Cond. U-Net”.**
>  In “Cond. U-Net” only the decoder is modulated while in “Fully Cond. U-Net” both decoder and encoder are modulated.

---

### Official Review · AnonReviewer2 · 2020-10-29
**An interesting idea and limited contribution**

**Rating:** 4
**Confidence:** 3

**Review:**

##########################################################################

Summary:

This paper propose a method for leveraging additional annotation by using an auxiliary network that modulates activations of the main network. The method proposed achieves significant improvements over a strong baseline on two datasets.

##########################################################################

Reasons for score:

ves:
+ This paper tackle the problem of out-of-distribution generalization which is crucial for machine learning applications.
+ The idea of leveraging additional information to enhance domain shift generalization is interesting and make sense to me.
+ The proposed conditional networks is practical.
+ Overall, the paper is well written and is easy to follow.

cons:
- My main concern about the paper is the novelty. The paper seems the incremental work based on Conditional Batch Normalization, which limits the contribution. Overall, the novelty and contribution of this work are marginal.

##########################################################################

---

> ### Author Response · Authors · 2020-11-23
> **Clarification of contribution**
>
> Thank you for your feedback and positive comments. We would like to point out that the conditional networks are not designed as an extension of conditional batch normalization. **Conditional Networks are a novel way to perform self-supervised learning where the pretext task exploits available metadata and it is learned jointly with the downstream task. Conditional computation is just one of the tools used to achieve it.** We are not aware of other work that uses CBN/CGN in the same way. In most cases, other modalities are fed as input to the network that generates the CBN parameters.

---

### Official Review · AnonReviewer1 · 2020-11-02
**Appreciated the proposed approach but was not convinced by the demonstration of the stated claims.**

**Rating:** 4
**Confidence:** 4

**Review:**

The paper intends to tackle the problem of out-of-distribution generalization through conditional computation.
The proposed framework combines conditional computation (exploiting extra information available about the problem data) with a task specific neural architecture.
Practically, a conditioning network is trained using image embeddings extracted from input images in order to predict extra annotations available about said images. It’s subsequently used to modulate layers of the main task network.
This type of Self-supervised learning in the framework will use available relevant context and embedded metadata as supervisory signals.
The main stated goal is the evaluation of how conditional networks improve generalization in the presence of distributional shift.
The experiments claim that the conditional network improves performance in certain tasks, namely semantic segmentation and image classification .
Experiments for these tasks are carried using Inria Aerial Image Labeling (aerial building segmentation task) and the Tumor-Infiltrating Lymphocytes (classification task).

Questions:
-	In 3 FORMULATION AND NETWORK ARCHITECTURE/ 3.1 PROBLEM ABSTRACTION: you state :” To encourage this, we train an auxiliary network to predict the …. Consider the example of building segmentation discussed in Section 4.1, where we use geographical coordinates as extra information. Imagine the model is presented with an image from a new city, which might have features that are visually similar to cities from the training set, but has very different coordinates. If we conditioned on the location alone, we would not be able to generalize to this new city.” This begs three questions:
-	Is this kind of example enough to justify using the intermediate high-level features ^z (xn) as a proxy for zn.?
-	In the given example, wouldn’t your framework equate to the main task network (No conditioning part used)?
-	Isn’t this testament to the fact that the conditioning addition to the main task network usefulness is highly dependent on the relevance of the extra information used to the main task?
-	In 4.1 CONDITIONAL NETWORKS FOR SEMANTIC SEGMENTATION OF AERIAL IMAGES: You state “The transfer set also has variation in illumination, landscape, and time, making it well-suited to evaluate out-of distribution generalization.” Can you clarify how those variations were obtained? Image selection for the second set of cities etc.
-	In 4.2/Methods tested: you state:  “the conditioning network is trained to perform image classification for the type of cancer (tn).” Can you give more detail on how that was done?

---

> ### Author Response · Authors · 2020-11-23
> **Improved description of method and motivation as well as experimental details**
>
> Thank you for your comments.
> Let us address your questions and concerns:
> 1. **Is this kind of example enough to justify using the intermediate high-level features ^z (xn) as a proxy for zn?**
> We have revised the text to clarify the motivation for the proposed method of implicit conditioning. To summarize the changes: **We removed the potentially confusing distinction between z_n and ^z_n.** Learning to capture visual characteristics that are correlated with the auxiliary targets allows the auxiliary network to predict (potentially) useful context for the main network, independent of the unknown values of t. Since the mapping from z to the conditional normalization parameters is also learned, the network can handle varying degrees of relevance of t for the main task.
> Besides requiring auxiliary information at test time, direct conditioning on t could harm the main networks performance, if values are outside of the ranges of auxiliary training data.
> 2. **In the aerial labeling example, wouldn’t your framework equate to the main task network (No conditioning part used)?**
>  Not really, the conditioning network plays an important role to learn adequate features z(xn). Not using the conditioning part is equivalent to “Cond. VGG16 alpha 0” shown in Table 3 for the TIL classification task. Table 3 results show the conditioning network is critical for conditional networks to work.
> 3. **Isn’t this testament to the fact that the conditioning addition to the main task network usefulness is highly dependent on the relevance of the extra information used to the main task?**
> Absolutely, and we have put more emphasis on this in our revision. *Please note that the proposed method can deal with varying degrees of relevance of the auxiliary data, since the mapping of z to the modulation parameters is also trained using the main objective.* The mapping can learn to simply output constant parameters that apply an identity transformation (all betas and gammas set to zero and one, respectively), which would *prevent negative transfer learning between unrelated tasks.*
> 4. **“The transfer set also has variation in illumination, landscape, and time, making it well-suited to evaluate out-of distribution generalization.” Can you clarify how those variations were obtained?**
> We use the previously released Inria image labeling dataset [1]. Authors purposely chose cities showing different landscapes to test out-of-distribution generalization. Imagery for the different cities was also collected in different times under different weather and illumination conditions.
> 5. **In 4.2/Methods tested: you state: “the conditioning network is trained to perform image classification for the type of cancer (tn).” Can you give more detail on how that was done?**
> The main task aims to determine whether the patient shows TIL cells or not (binary classification) in cancer patients. The dataset contains patients suffering from different cancer types and metadata indicating it is available. The conditioning task *(predicting t)* performed by the conditioning network is to classify the cancer type.
>
> **References**
> - 1. Emmanuel Maggiori, Yuliya Tarabalka, Guillaume Charpiat and Pierre Alliez. “Can Semantic Labeling Methods Generalize to Any City? The Inria Aerial Image Labeling Benchmark”. IEEE International Geoscience and Remote Sensing Symposium (IGARSS). 2017.

---

### Author Response · Authors · 2020-11-23
**Revised manuscript and rebuttal**

We thank all reviewers for their constructive feedback. We have uploaded a revised manuscript that applies suggested changes. The revised document better highlights our contribution to address concerns raised regarding the novelty over conditional batch normalization.
We also added details on the experimental setup, additional comments on the interpretation of conditioning features, clarified the intuition behind the proposed method, and fixed presentation issues. \
We responded to raised issues directly under the corresponding reviews.

---

### Decision · Program_Chairs · 2021-01-07
**Final Decision**

**Decision:**

Reject

**Comment:**

The paper proposes to address the out-of-distribution generalization problem by means of conditional computation in form of a feature modulating module.
While the approach is interesting and brings a new take on how to perform feature modulation (although initially felt too similar to Conditional Batch Normalization) some major concerns about the experiments and validation of the approach are raised by all reviewers. Some of the hypothesis made are also challenged due to lack of proper validation.
Although the discussion clarified some points I am afraid many open questions are left unanswered and would require a more work to be fully addressed before acceptance.